# Electrical Stimuli-Responsive Decomposition of Layer-by-Layer Films Composed of Polycations and TEMPO-Modified Poly(acrylic acid)

**DOI:** 10.3390/polym14245349

**Published:** 2022-12-07

**Authors:** Kentaro Yoshida, Toshio Kamijo, Tetsuya Ono, Takenori Dairaku, Shigehiro Takahashi, Yoshitomo Kashiwagi, Katsuhiko Sato

**Affiliations:** 1School of Pharmaceutical Sciences, Ohu University, 31-1 Misumido, Tomita-machi, Koriyama 963-8611, Fukushima, Japan; 2Department of Creative Engineering, National Institute of Technology, Tsuruoka College, 104 Sawada, Inooka, Tsuruoka 997-8511, Yamagata, Japan; 3Integrated Center for Science and Humanities, The Section of Chemistry, Fukushima Medical University, 1 Hikarigaoka, Fukushima City 960-1295, Fukushima, Japan; 4Faculty of Pharmacy, Takasaki University of Health and Welfare, 37-1 Nakaorui-cho, Takasaki 370-0033, Gunma, Japan; 5Faculty of Pharmaceutical Science, Tohoku Medical and Pharmaceutical University, 4-4-1 Komatsushima, Aoba, Sendai 981-8558, Miyagi, Japan

**Keywords:** layer-by-layer, multilayer film, thin layer film, electro-decomposition, DDS, TEMPO

## Abstract

We previously reported that layer-by-layer (LbL) film prepared by a combination of 2,2,6,6-tetramethylpiperidinyl *N*-oxyl (TEMPO)-modified polyacrylic acid (PAA) and polyethyleneimine (PEI) were decomposed by application of an electric potential. However, there have been no reports yet for other polycationic species. In this study, LbL films were prepared by combining various polycationics (PEI, poly(allylamine hydrochloride) (PAH), poly(diallydimethylammonium chloride) (PDDA), and polyamidoamine (PAMAM) dendrimer) and TEMPO-PAA, and the decomposition of the thin films was evaluated using cyclic voltammetry (CV) and constant potential using an electrochemical quartz crystal microbalance (eQCM). When a potential was applied to an electrode coated on an LbL thin film of polycations and TEMPO-PAA, an oxidation potential peak (Ep^a^) was obtained around +0.6 V vs. Ag/AgCl in CV measurements. EQCM measurements showed the decomposition of the LbL films at voltages near the Ep^a^ of the TEMPO residues. Decomposition rate was 82% for the (PEI/TEMPO-PAA)_5_ film, 52% for the (PAH/TEMPO-PAA)_5_ film, and 49% for the (PDDA/TEMPO-PAA)_5_ film. It is considered that the oxoammonium ion has a positive charge, and the LbL films were decomposed due to electrostatic repulsion with the polycations (PEI, PAH, and PDDA). These LbL films may lead to applications in drug release by electrical stimulation. On the other hand, the CV of the (PAMAM/TEMPO-PAA)_5_ film did not decompose. It is possible that the decomposition of the thin film is not promoted, probably because the amount of TEMPO-PAA absorbed is small.

## 1. Introduction

Stimuli-sensitive systems such as gels, thin films, and microcapsules have been studied for their various applications in fields including biomaterials, energy storage and conversion. To achieve such a functional application, techniques to combine predictable responses to stimuli-materials, and controlled composition and nanostructural have been reported [1,2]. A diversity of polyelectrolytes has been used in the research on many nanostructured systems. As stimulus-responsive polymers, synthetic polymers modified with functional substances and enzymes of biomaterials have been reported in fields such as biology, organics, and polymer chemistry [3,4,5]. In addition, layer-by-layer (LbL) deposition is attracting attention as a combination of functional polymers and nanostructures. Functional polymers for LbL films include synthetic polymers, polysaccharides, proteins, and DNA [6,7,8]. Thin films can have various functions through the use of such functional polymers. In particular, LbL films have been used as stimuli-responsive thin films for stimuli such as pH [9], ionic strength [10], electrochemical stimulus [11,12], temperature [13], sugars [14,15], and hydrogen peroxide [16]. In the case of pH and ionic strength stimuli-responsive thin films, the charge density of the weak polyelectrolyte functional species changes with the pH and ionic strength of the solvent. In particular, thin films composed of amphoteric polyelectrolytes with weak positive and negative charges can be decomposed by a pH-trigger [17]. We have recently prepared LbL films by combining glucose oxidase, DNA, and hemin, and reported thin film decomposition in response to sugar and hydrogen peroxide [18,19,20].

In the case of electrical stimuli-responsive thin films, there are reports of the use of redox polymers and indefinite nanoparticles. For example, ferrocene-modified polymers can be decomposed by the application of an oxidizing potential in thin films based on the redox-active components of LbL films [21]. In addition, redox active substances such as alizarin red, polypyrrole, and metal nanoparticles are also used in LbL films with redox activity [22,23,24]. Electrostimuli-responsive films are expected to be applied to biosensors and drug delivery systems (DDS), such as with the electrochemical decomposition/swelling of thin films.

Our study is focused on 2,2,6,6-tetramethylpiperidinyl *N*-oxyl (TEMPO) derivatives. The TEMPO derivative is a stable nitroxide radical that is often used as a catalyst for various oxidative transformations (alcohol oxidation, oxygen reduction, and various organic reactions [25,26,27]). One-electron oxidation of nitroxides produces oxoammonium cations that provide active oxidants (Figure 1) [28].

The oxoammonium cation can be used in stoichiometric quantities or generated electrochemically in situ, or through the use of a variety of secondary oxidants [29]. Therefore, in the case of LbL films driven by electrostatic interactions, the stability of thin films composed of TEMPO derivatives should be dependent on the electrode potential. We previously reported that LbL films prepared by a combination of TEMPO-modified polyacrylic acid (PAA) and polyethyleneimine (PEI) were decomposed by application of an electric potential [30]. This is due to the oxidation of the TEMPO derivative, and the increase in the positive charge density of the weak polyelectrolyte degrades the balance of the thin film driven by the electrostatic interaction, which leads to decomposition of the LbL film (Figure 2).

However, there have been no reports yet for other polycationic species. Elucidating the positive charge state of primary and quaternary amines and the degradability of thin films in linear and branched polymers may lead to applications in drug release upon electrical stimulation. In this study, LbL membranes were prepared by combining various polycationics (PEI, poly(allylamine hydrochloride) (PAH), poly(diallydimethylammonium chloride) (PDDA), and polyamidoamine (PAMAM) dendrimer) and TEMPO-PAA, and the decomposition of the thin films was evaluated using cyclic voltammetry (CV) and constant potential using an electrochemical quartz crystal microbalance (eQCM). Furthermore, the fluorescence intensity of solutions in which the thin films were degraded was measured using fluorescein isothiocyanate (FITC)-labeled polycations as a drug model compound. 

## 2. Materials and Methods

### 2.1. Materials

Poly(acryloyl chloride) (25% soln. in dioxane) was purchased from Polysciences Inc. (Warrington, PA, USA). 4-Amino-2,2,6,6,-tetramethylpiperidine *N*-oxyl free radical (4-amino-TEMPO) was purchased from Tokyo Chemical Inc. (Tokyo, Japan). PAH (average MW: 150,000) and PEI (MW: 60,000–80,000) were obtained from Nittbo Co. (Tokyo, Japan) and Nacalai Tesque Inc. (Tokyo, Japan)). PDDA (MW: 100,000–200,000), PAMAM dendrimer (ethylenediamine core, generation 4.0), and FITC isomer I were obtained from Sigma-Aldrich, Co (Saint Louis, MO, USA).

TEMPO-PAA was synthesized as follows. Poly(acryloyl chloride) (100 mg), 4-amino-TEMPO (80 mg), and triethylamine (80 μL) were mixed in 5 mL of dioxane for 24 h. Ten milliliters of water was added to the reaction mixture to and stirred for 24 h. The product was purified by dialysis with milliQ water for 3 days. The TEMPO-PAA contained approximately 35% 4-amino TEMPOresidues as calculated from the proportions of nitrogen and carbon determined by elemental analysis (C, 55.67%; H, 8.31%; N, 7.43%). The chemical structures of TEMPO-PAA, PEI, PAH, PAMAM, and PDDA are shown in Figure 3. FITC-labeled polycations (FITC-PEI, FITC-PAH, and FITC-PAMAM) were synthesized as follows. The various polycations (100 mg) and FITC (amount that modifies 1% of primary amines contained in various polyanions) were mixed in an aqueous solution at 4 °C for over 12 h. The product was purified by dialysis with milliQ water for 3 days.

### 2.2. Analyses

A quartz crystal microbalance (QCM; QCA 917 system, Seiko EG&G, Tokyo, Japan) was employed for gravimetric analysis of the dried LbL films consisting of polycations and TEMPO-PAA. A 9 MHz AT-cut quartz resonator coated with a gold (Au) layer (surface area 0.4 cm^2^) was used as a probe. An eQCM (eQCM 10M, Gamry, Warminster, UK) was used for gravimetric analysis of the wet LbL films. An 8 MHz AT-cut quartz resonator coated with a thin Au layer (0.2 cm^2^) was used as a probe. Atomic force microscopy (AFM; AFM5200S, Hitachi High-Technologies Co., Tokyo, Japan) images were acquired in contact mode at room temperature and in air. An electrochemical analyzer (ALS model 1200B, BAS Inc., Tokyo, Japan) was used for cyclic voltammetry and application of a constant potential. Fluorescence spectroscopy measurements were conducted using a fluorescence spectrophotometer (RF-1500, Shimadzu Co., Kyoto, Japan). UV-vis spectroscopy measurements were conducted using a UV-3100PC (Shimadzu Co., Kyoto, Japan) spectrometer.

### 2.3. Preparation of LbL Films

LbL films were prepared on a solid substrate (Au-coated quartz resonator for eQCM analysis), quartz slide (50 × 9 × 1 mm) and glassy carbon electrode (GCE) slides for constant potential application. The solid substrates were alternately immersed in 0.1 mg/mL polycation solution and 0.1 mg/mL TEMPO-PAA solution for 15 min. The slides were rinsed with a working buffer for 5 min to remove any weakly adsorbed polycations and TEMPO-PAA. The deposition was repeated to build up LbL films. All experiments were conducted at room temperature.

Deposition was repeated to construct the LbL films on a 9-MHz Au-coated quartz resonator (surface are 0.4 cm^2^), after which the films were rinsed with water, and the change in the resonance frequency of the dry film was recorded after each deposition to determine its weight. In addition, the surface morphology of the dry film was recorded using AFM in contact mode.

### 2.4. Cyclic Voltammetry and Constant Potential Application

Cyclic voltammograms (CVs) and eQCM (8-MHz Au-coated quartz resonator, surface are 0.2 cm^2^) were recorded in a conventional three-electrode system using a stainless steel tube counter electrode and Ag/AgCl reference electrode (eQCM flow cell kit, BAS Inc., Tokyo, Japan). All measurements were performed. The LbL film-coated Au quartz resonator was immersed in 10 mM HEPES buffer (pH 7.0, 150 mM NaCl) for 10 min. The electrode potential of the Au-coated resonator was scanned from +0 to +0.8 V at 0.05 V/s, and constant potentials of 0, +0.4, +0.6, and +0.8 V were applied for 10 min. The eQCM was measured while the potential was applied to the quartz resonator.

### 2.5. Release of FITC-Polycation Due to Constant Potential Application Obtained from the (FITC-Polycation/TEMPO-PAA)_5_ Films-Coated Electrodes

To release FITC-polycations (FITC-PEI, FITC-PAH, and FITC-PAMAM), electrode potentials from +0 to +0.8 V were applied to the (FITC-polycation/TEMPO-PAA)_5_ film-coated GCEs (50 × 9 × 1 mm) in 10 mM HEPES buffer (pH 7.0, 150 mM NaCl). The constant potential of the film-coated GCEs was applied in a conventional three-electrode system using a platinum wire counter electrode and an Ag/AgCl reference electrode.

The fluorescence intensity of the FITC-polycations released from the LbL films was determined at 520 nm using a fluorescence spectrophotometer with an excitation wavelength of 490 nm. Release of the FITC-polycations was calculated based on the fluorescence intensity of the solution in which a potential was applied to the LbL film-coated GCEs for a predetermined time. A calibration graph obtained using the fluorescence intensity of free FITC-polycations was used.

## 3. Results

### 3.1. Preparation of (Polycation/TEMPO-PAA)_n_ Films

We have previously reported that multilayer films can be formed by LbL deposition utilizing the electrostatic interaction between polymers [31]. Here, we evaluate whether or not polycations and TEMPO-PAA also form multilayer films. Figure 4 shows the change in the resonance frequency (ΔF) of QCM measurements observed during the deposition of polycations and TEMPO-PAA on a Au-coated 9 MHz quartz resonator (surface are: 0.4 cm^2^). When the thin film is dried, the frequency change of the probe can be obtained from the Sauerbrey equation to determine the mass deposited onto the quartz resonator; the deposition of 1 ng of a substance induces a −0.91 Hz change in the resonance frequency. The amount of deposited LbL film was calculated to be ca. 3.41 ± 0.54 μg/cm^2^ for the (PAH/TEMPO-PAA)_5_ film, ca. 3.16 ± 0.21 μg/cm^2^ for the (PEI/TEMPO-PAA)_5_ film, ca. 2.13 ± 0.29 μg/cm^2^ for the (PAMAM/TEMPO-PAA)_5_ film, and ca. 2.02 ± 0.16 μg/cm^2^ for the (PDDA/TEMPO-PAA)_5_ film. The difference in the film formation of each polycation will be described later; however, the formation of LbL films with TEMPO-PAA was possible for all polycations.

To further verify the LbL film preparation, AFM observations of dried (PEI/TEMPO-PAA)_5_, (PAH/TEMPO-PAA)_5_, (PAMAM/TEMPO-PAA)_5_, and (PDDA/TEMPO-PAA)_5_ films were conducted (Figure 5). The roughness values ranged from 68.0 nm for the (PEI/TEMPO-PAA)_5_ film, 65.7 nm for the (PAH/TEMPO-PAA)_5_ film, 61.7 nm for the (PAMAM/TEMPO-PAA)_5_ film, and 9.19 nm for the (PDDA/TEMPO-PAA)_5_ film. All the obtained LbL films have TEMPO-PAA as the outermost layer, and the surface morphology was similar. Many examples of LbL films surface roughness on the scale of several tens of nm have been reported [18,30]. These results are presumed to be similar to the adsorption of polymers in the LbL films. 

### 3.2. Electrical Stimuli-Responsive Decomposition of (Polycation/TEMPO-PAA)_5_ Film-Coated Electrodes

The dried (PEI/TEMPO-PAA)_5_ film was immersed in the buffer solution and a potential was applied using CV. However, the frequency change in the dry QCM measurement and the surface state in the AFM observation did not change. For dried LbL films, the AFM thickness values are known to be much lower than those obtained for wet LbL films [32,33]. This may be due to the increase in the density of the thin film, which also made it difficult to decompose the thin film. Therefore, in all subsequent experiments, the decomposition of the thin film, which is possible in solution, was investigated by application of an electric potential using a flow-type QCM.

#### 3.2.1. CVs and eQCM Measurements Obtained from (PEI/TEMPO-PAA)_5_ Film-Coated Electrodes

Figure 6 shows CVs and eQCM measurements obtained for the (PEI/TEMPO-PAA)_5_ film-modified quartz resonator. The CVs contain an oxidation peak at +0.6 V due to the oxidation reaction of TEMPO in the (PEI/TEMPO-PAA)_5_ film. An oxidation current peak (*Ip^a^*) of 74 μA was observed at an oxidation potential peak (*Ep^a^*) of +0.64 V in the 1st sweep. However, a significant decrease in *Ip^a^* was observed after the 2nd sweep. The corresponding eQCM measurement showed a sharp increase in ΔF around +0.6 V during the 1st sweep. ΔF after the sweep scan showed an increase of approximately +690 Hz. The increased ΔF shows that approximately 82% of the weight of the (PEI/TEMPO-PAA)_5_ film was removed from the Au-coated quartz resonator because the adsorption of the LbL film caused a change in ΔF of −842 Hz. From the eQCM changes after the 2nd sweep, it can be inferred that all the TEMPO-PAA that could be resolved in the 1st sweep was exfoliated from the quartz resonator (Note: the layer of polyelectrolyte directly adsorbed on the substrate is not exfoliated).

The mechanism of LbL film decomposition is as follows. The TEMPO residues were electrochemically oxidized at +0.6 to +0.7 V to form positively charged oxoammonium ions. We have reported that the decomposition of the film is probably caused by electrostatic repulsion between the positively charged oxoammonium ions and PEI [30]. Sodium hypochlorite is used to oxidize TEMPO. The absorbance spectra of the LbL films decreased significantly when the (PEI/TEMPO-PAA) films were immersed in sodium hypochlorite solution (Appendix A). It is assumed that the LbL film decomposed due to the oxidation of the TEMPO derivative.

The usefulness of the LbL film composed of polycations with covalently modified TEMPO is discussed here. Figure 7 shows CVs and eQCM measurements obtained from the (PEI/PAA)_5_ film-modified Au electrode in 10 mM TEMPO solution. The CVs indicated a reversible redox response of TEMPO for 50 sweep scans. However, no increase in ΔF was observed in the eQCM measurements, i.e., the LbL film does not decompose. PAA with covalently modified TEMPO increases the intramolecular positive charge depending on the oxidation potential. The change in the charge of the polymer in the thin film disturbs the balance of the electrostatic interaction, which is the driving force that constitutes the LbL films, and this causes the LbL film to decompose. 

#### 3.2.2. CVs and eQCM Measurements Obtained from (PAH/TEMPO-PAA)_5_ Film-Coated Electrodes

Figure 8 shows CVs and eQCM measurements obtained for the (PAH/TEMPO-PAA)_5_ film-modified Au electrode. The CVs show *Ep^a^* at +0.7 V due to the oxidation reaction of TEMPO in the (PAH/TEMPO-PAA)_5_ film. At *Ep^a^* in the 1st scan, ΔF in the eQCM measurement increased due to the oxidation reaction of the (PEI/TEMPO-PAA)_5_ film. Similar to the (PAH/TEMPO-PAA)_5_ film, the increase in ΔF involves the decomposition of the (PAH/TEMPO-PAA)_5_ film. The *Ip^a^* in the CV after the 2nd scan decreased significantly compared to that of the 1st scan, and the rate of change with the increase in ΔF decreased. After the 5th scan, ΔF showed an increase of approximately +2450 Hz. After an additional 50 scans, ΔF showed a total increase of ca. +2880 Hz. The adsorption of the (PAH/TEMPO-PAA)_5_ film caused a change in ΔF of −5630 Hz, i.e., approximately 52% of weight of the (PAH/TEMPO-PAA)_5_ film was removed from the Au-coated quartz resonator. The (PAH/TEMPO-PAA)_5_ film thus decomposes similarly to the (PEI/TEMPO-PAA)_5_ film.

#### 3.2.3. CVs and eQCM Measurements Obtained from (PDDA/TEMPO-PAA)_5_ Film-Coated Electrodes

PDDA is a high charge density cationic polymer with quaternary amines. Figure 9 shows CVs and eQCM measurements obtained from the (PDDA/TEMPO-PAA)_5_ film-coated Au electrode. The CVs show *Ep^a^* at +0.6 V due to the oxidation reaction of TEMPO in the (PDDA/TEMPO-PAA)_5_ film. At *Ep^a^* in the 1st scan, ΔF from the eQCM measurements increased due to the oxidation reaction of the (PDDA/TEMPO-PAA)_5_ film. In the 2nd and subsequent scans, *Ip^a^* was significantly decreased and there was no change in ΔF compared to that of the 1st scan. This shows a tendency similar to that of the (PEI/TEMPO-PAA)_5_ film. After the 55th scan, ΔF showed an increase of approximately +208 Hz. Figure 9B shows that approximately 49% of the mass of the (PDDA/TEMPO-PAA)_5_ film was removed from the Au-coated quartz resonator because the adsorption of the (PDDA/TEMPO-PAA)_5_ film caused a change in ΔF of −421 Hz.

#### 3.2.4. CVs and eQCM Measurements Obtained from (PAMAM/TEMPO-PAA)_5_ Film-Coated Electrodes

PAMAM is a spherical macromolecule that can sequester low molecular weight compounds in cavities, and PAMAM has been extensively studied as a drug carrier for DDS [34,35]. If the membrane that includes PAMAM can be decomposed electrochemically, it may be applicable to DDS. Figure 10 shows CVs and eQCM measurements obtained from the (PAMAM/TEMPO-PAA)_5_ film-modified Au electrode. The CVs show *Ep^a^* at +0.6 V due to the oxidation reaction of TEMPO in the (PAMAM/TEMPO-PAA)_5_ film during the 1st scan. The adsorption of the (PAMAM/TEMPO-PAA)_5_ film caused a change in ΔF of −710 Hz, which was approximately the same amount of change in ΔF as the (PEI/TEMPO-PAA)_5_ film. However, *Ip^a^* in the 1st sweep scan of the (PAMAM/TEMPO-PAA)_5_ film was ca. 7 μA, which is significantly smaller than that of the (PEI/TEMPO-PAA)_5_ film. Therefore, even if the amount of change in ΔF is the same, the amount of adsorption of TEMPO-PAA is small. Alternatively, the film may be in a state where it cannot be oxidized and reduced.

#### 3.2.5. Changes in Resonance Frequency Due to Application of a Constant Potential

The decomposition of LbL films at constant potential was investigated. Figure 11 shows the changes in the resonance frequency upon application of +0.4, +0.6, and +0.8 V to the (PEI/TEMPO-PAA)_5_, (PAH/TEMPO-PAA)_5_, (PDDA/TEMPO-PAA)_5_, and (PAMAM/TEMPO-PAA)_5_ film-coated electrodes. In the electrode coated with the (PEI/TEMPO-PAA)_5_ and (PAH/TEMPO-PAA)_5_ films, the change in ΔF was slight when a potential of +0.4 V was applied, but an increase in ΔF was observed when a potential greater than +0.6 V was applied. The decomposition rate was calculated from the amount of change in ΔF, which indicated the decomposition of the (PEI/TEMPO-PAA)_5_ and (PAH/TEMPO-PAA)_5_ films was ca. 50% and 30% at +0.6 V, and ca. 90% and 79% at +0.8 V, respectively. Decomposition of the (PDDA/TEMPO-PAA)_5_ film was also observed at +0.8 V. On the other hand, the PAMAM film that was not decomposed during the CV measurements was not decomposed even by application of a constant potential of +0.8 V.

### 3.3. Release of FITC-Modified Polycations under Constant Potential

LbL films can contain drugs or can be made from polymer electrolytes such as insulin [36,37]. By skillfully using TEMPO derivatives, LbL films could be decomposed by electrical stimulation; therefore, we have investigated the possibility of application as a DDS. The polycations were labeled with FITC as a release model compound, and the release of FITC-polycations was evaluated from the fluorescence intensity of the solution in which the LbL films were electrically stimulated. Figure 12 shows the release of FITC-polycations upon application of +0, +0.4, +0.6, and +0.8 V to the (FITC-PEI/TEMPO-PAA)_5_, (FITC-PAH/TEMPO-PAA)_5_, and (FITC-PAMAM/TEMPO-PAA)_5_ film-coated GCEs. FITC-PEI was released from the (FITC-PEI/TEMPO-PAA)_5_ film by the application of +0.6 V for 10 min. In the (FITC-PEI/TEMPO-PAA)_5_ film, the amount of FITC-PEI released was increased with the potential from +0.4 V. Similar to the decomposition behavior of LbL films under low potential application, the release characteristics of FITC-PEI could also be controlled by manipulating the potential. However, the amount of FITC-PEI released was decreased when a potential of +0.8 V was applied. This is because FITC is oxidized to a non-fluorescent substance by the application of a potential [38]. If a DDS that is triggered by electric potential stimulation could be constructed, it would be necessary to do so at an electric potential at which the target drug does not undergo an electrochemical reaction. On the other hand, the (FITC-PAH/TEMPO-PAA)_5_ and (FITC-PAMAM/TEMPO-PAA)_5_ films did not show efficient decomposition by application of an electric potential. Although the amount of the (PAH/TEMPO-PAA)_5_ film that was absorbed was larger than that of the (PEI/TEMPO-PAA)_5_ film, the amount of FITC-PAH released was significantly smaller. Although the reason for this is still under investigation, it is speculated that the functional groups of PAHs are all primary amines and that they reacted with FITC upon application of the electric potential. Similar to the eQCM results, there was no release of FITC-PAMAM.

## 4. Conclusions

*Ep^a^* was observed around +0.6 V in CV measurements when a potential was applied to the (polycation/TEMPO-PAA) film-coated quartz resonator. This was due to the electrochemical oxidation of the TEMPO derivatives to generate oxoammonium ions [29]. At the same time, decomposition of the LbL films was observed at *Ep^a^* around +0.6 V due to a sharp increase in ΔF. It is considered that the oxoammonium ion has a positive charge, and the LbL films were decomposed due to electrostatic repulsion with the polycations (PEI, PAH, and PDDA). On the other hand, the CV of the (PAMAM/TEMPO-PAA)_5_ film showed a small *Ip^a^* during the 1st sweep scan, and the LbL film did not decompose. It is possible that the decomposition of the thin film is not promoted, probably because the amount of TEMPO-PAA absorbed is small. Furthermore, in the LbL films to which a constant potential was applied, decomposition was accelerated as the potential approached the *Ep^a^* of TEMPO. However, when a constant potential of +0.8 V was applied, the release of FITC-modified polycations from the thin film was suppressed. It is possible that FITC and primary amines were electrochemically oxidized. If a DDS triggered by electric potential stimulation could be constructed, it would be necessary to do so at an electric potential at which the target drug does not undergo an electrochemical reaction. If the target drug does not cause an electrochemical reaction, it may be possible to construct DDS by electric potential stimulation.

## Figures and Tables

**Figure 1 polymers-14-05349-f001:**
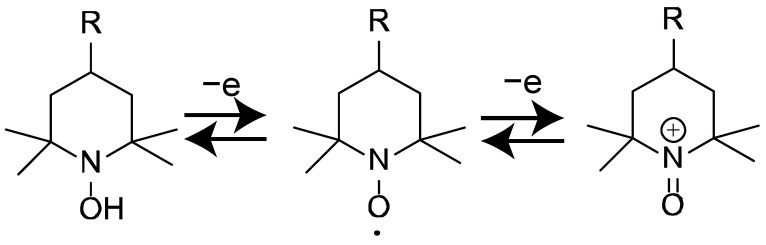
Reversible redox system based on nitroxyl radicals.

**Figure 2 polymers-14-05349-f002:**
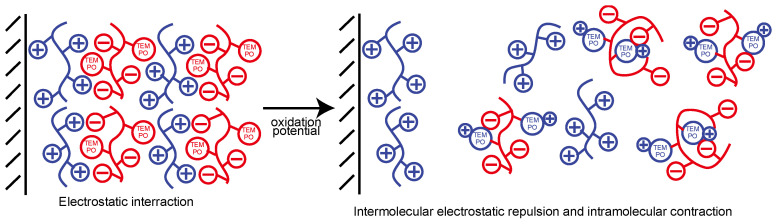
Response of TEMPO-PAA and polycations to electrical-stimulation. Schematic depiction of the molecular and conformational responses of polyelectrolytes bearing either basic, acid, or TEMPO moieties with respect to electrical-stimuli.

**Figure 3 polymers-14-05349-f003:**
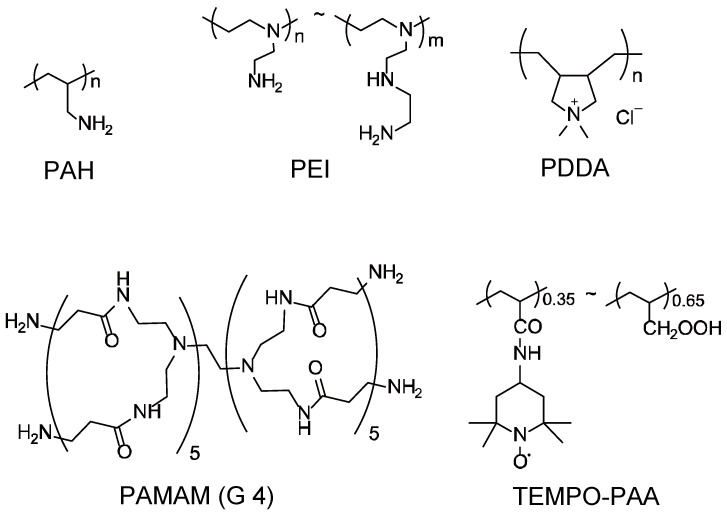
Chemical structures of PAH, PEI, PDDA, PAMAM and TEMPO-PAA.

**Figure 4 polymers-14-05349-f004:**
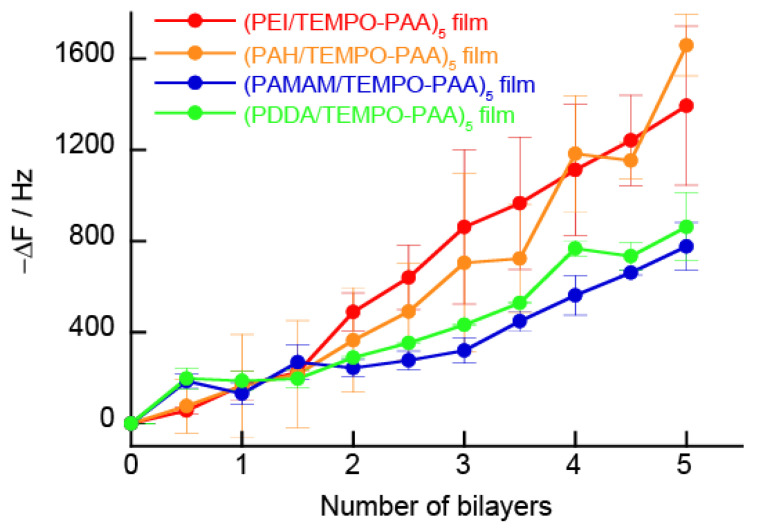
QCM resonator frequency changes during drying of the prepared (PEI/TEMPO-PAA)_n_ (red), (PAH/TEMPO-PAA)_n_ (orange), (PAMAM/TEMPO-PAA)_n_ (blue), and (PDDA/TEMPO-PAA)_n_ (green) films. Surface area: 0.4 cm^2^.

**Figure 5 polymers-14-05349-f005:**
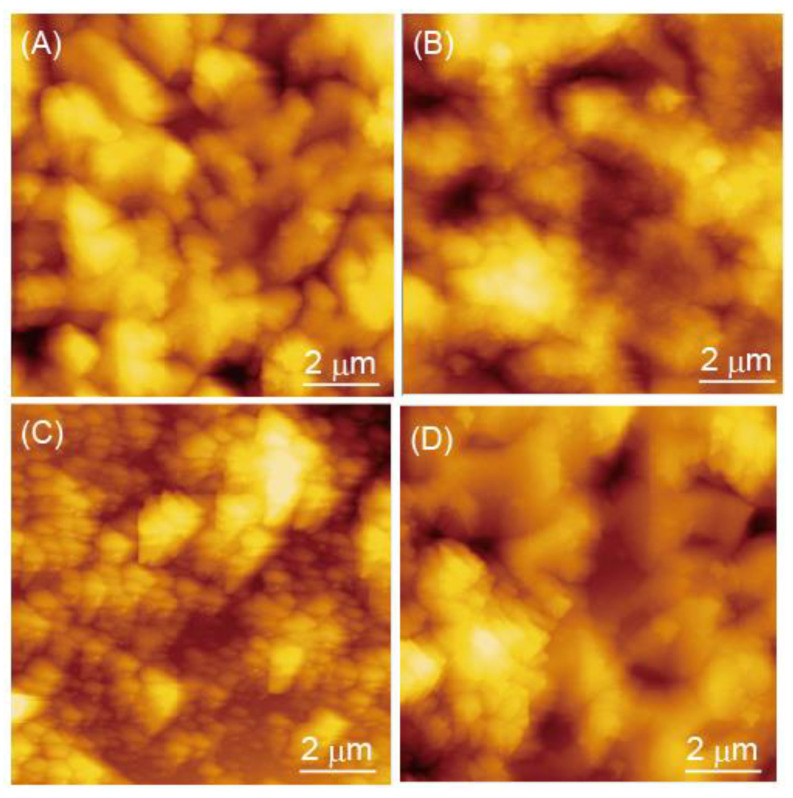
AFM images of dried (**A**) (PEI/TEMPO-PAA)_5_, (**B**) (PAH/TEMPO-PAA)_5_, (**C**) (PAMAM/TEMPO-PAA)_5_, and (**D**) (PDDA/TEMPO-PAA)_5_ films.

**Figure 6 polymers-14-05349-f006:**
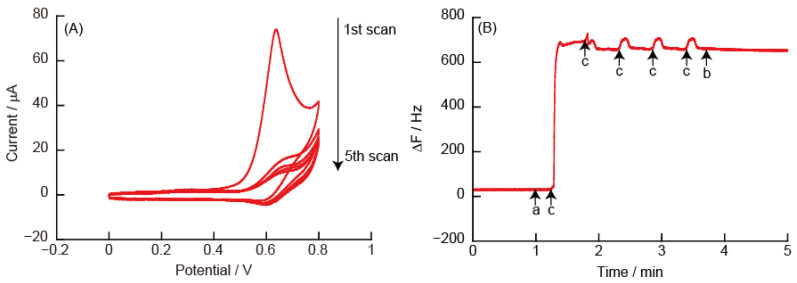
(**A**) Cyclic voltammograms of (PEI/TEMPO-PAA)_5_ film in 10 mM HEPES (pH 7, containing 150 mM NaCl), and (**B**) changes in the resonance frequency on the Au resonator in eQCM. The electric potential (scan rate: 50 mV/s, number of sweep scan: 5 cycles) was applied between (a) and (b) in the eQCM, and (c) indicates an oxidation potential of +0.6 V.

**Figure 7 polymers-14-05349-f007:**
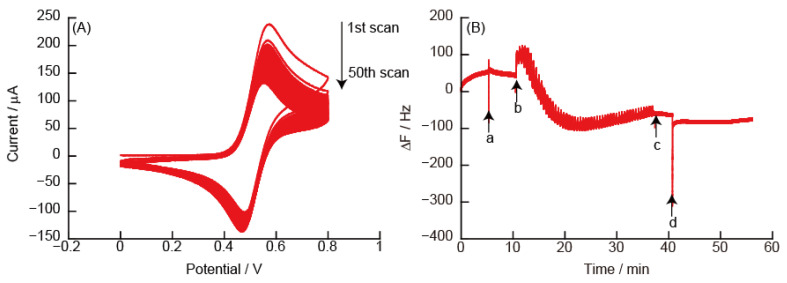
(**A**) Cyclic voltammograms of (PEI/PAA)_5_ film in 10 mM TEMPO solution, and (**B**) changes in the resonance frequency on the Au resonator in the eQCM. During QCM measurements, the Au resonator was exposed in (a) 10 mM TEMPO solution and (d) working buffer. The electric potential (scan rate: 50 mV/s, number of sweep scan: 5 cycles) was applied between (b) and (c) in the eQCM.

**Figure 8 polymers-14-05349-f008:**
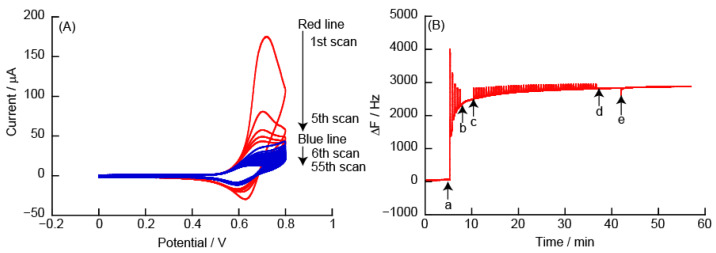
(**A**) Cyclic voltammograms of (PAH/TEMPO-PAA)_5_ film in 10 mM HEPES (pH 7, containing 150 mM NaCl), and (**B**) changes in the resonance frequency on the Au resonator in the eQCM. The electric potential (scan rate: 50 mV/s) was applied between (a) and (b) (number of sweep scan: 5 times), or (c) and (d) (number of sweep scan: 50 times) in the eQCM. Finally, the Au resonator was exposed to 10 mM HEPES buffer (pH 7, containing 150 mM NaCl) (e).

**Figure 9 polymers-14-05349-f009:**
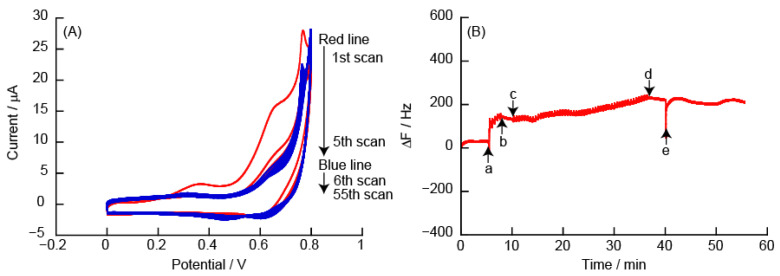
(**A**) Cyclic voltammograms of (PDDA/TEMPO-PAA)_5_ film in 10 mM HEPES (pH 7, containing 150 mM NaCl), and (**B**) changes in the resonance frequency on the Au resonator in eQCM. The electric potential (scan rate: 50 mV/s) was applied between (a) and (b) (number of sweep scan: 5 times), or (c) and (d) (number of sweep scan: 50 times) in the eQCM. Finally, the Au resonator was exposed to 10 mM HEPES buffer (pH 7, containing 150 mM NaCl) (e).

**Figure 10 polymers-14-05349-f010:**
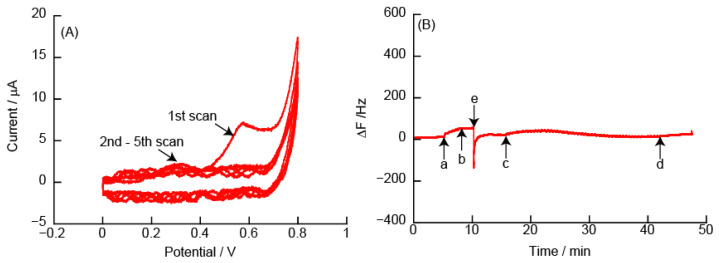
(**A**) Cyclic voltammograms of (PAMAM/TEMPO-PAA)_5_ film in 10 mM HEPES (pH 7, containing 150 mM NaCl), and (**B**) changes in the resonance frequency on the Au resonator. The electric potential (scan rate: 50 mV/s) was applied between (a) and (b) (number of sweep scan: 5 times), or (c) and (d) (number of sweep scan: 50 times) in the eQCM. The Au resonator was exposed to 10 mM HEPES buffer (pH 7, containing 150 mM NaCl) (e).

**Figure 11 polymers-14-05349-f011:**
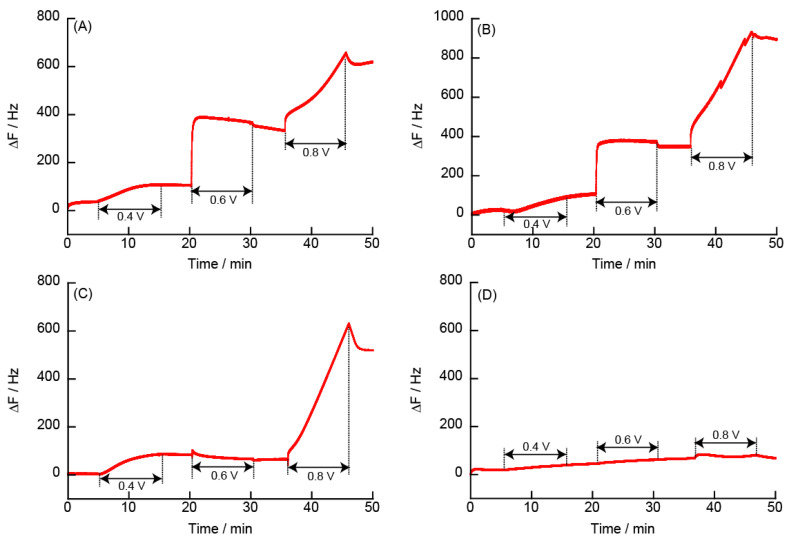
Changes in the resonance frequency for the (**A**) (PEI/TEMPO-PAA)_5_, (**B**) (PAH/TEMPO-PAA)_5_, (**C**) (PDDA/TEMPO-PAA)_5_, and (**D**) (PAMAM/TEMPO-PAA)_5_ films with coating electrodes and constant potential. The electrode potential was applied at +0.4 V, +0.6 V, and +0.8 V for 10 min.

**Figure 12 polymers-14-05349-f012:**
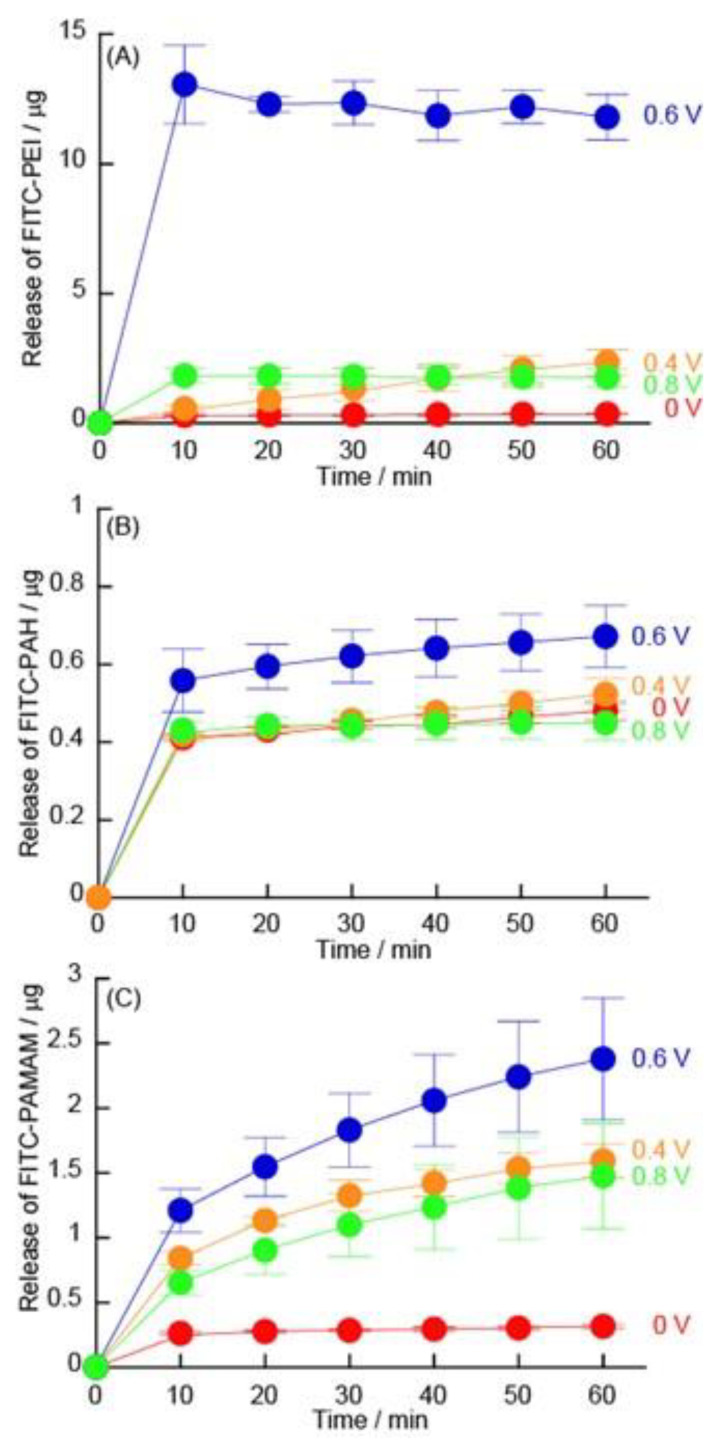
Release of (**A**) FITC-PEI from (FITC-PEI/TEMPO-PAA)_5_, (**B**) FITC-PAH from (FITC-PAH/TEMPO-PAA)_5_ film, and (**C**) FITC-PAMAM from (FITC-PAMAM/TEMPO-PAA)_5_ films coated on GCE at different constant potentials (red: 0 V, orange: +0.4 V, blue: +0.6 V, and green: +0.8 V).

## Data Availability

Not applicable.

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
