# Peer review of "Electrical Stimuli-Responsive Decomposition of Layer-by-Layer Films Composed of Polycations and TEMPO-Modified Poly(acrylic acid)"

_polymers, 2022, doi:10.3390/polym14245349_

Round 1
Reviewer 1 Report
In this manuscript, the authors prepared electrical stimuli-responsive thin films composed of polycations and 2,2,6,6-tetra-methylpiperidinyl N-oxyl modified polyacrylic acid (TEMPO-PAA) by layer-by-layer (LbL) film deposition. I hope the author can respond to the following questions:
1. What are the novelties of this work with respect to the previously published works? For example, what is the advantages of electrical stimuli? It is necessary to exactly and compressively explain the novelties of this work within the introduction section.
2. The abstract is tedious, it should be reorganized.
3. The formation of positively charged oxoammonium ions and the mechanism of LbL film decomposition need data support.
4. Please check that if the text (line 170) on page 5 is complete. “When the thin film is dried, the frequency change of the probe can be obtained fro”
Author Response
Thank you for your kind suggestion of revision of our manuscript (polymers- 2056304).We have revised the manuscript according to reviewer’s comments. All revisions made are marked in red in the revised manuscript. Our responses are as follows.
1.What are the novelties of this work with respect to the previously published works?For example, what is the advantages of electrical stimuli? It is necessary to exactly and compressively explain the novelties of this work within the introduction section.
→Added text (in red) to the introduction (page.3). We investigated whether the degradability of thin films differs depending on the functional groups of various amines and differences in the polymer structure.
2.The abstract is tedious, it should be reorganized.
→Revised abstract.
3.The formation of positively charged oxoammonium ions and the mechanism of LbL film decomposition need data support.
→As an example of oxidizing TEMPO derivatives, sodium hypochlorite is used as an oxidizing agent. Sodium hypochlorite immersion also decomposed the LbL film (Figure S1). Added text to the manuscript(page 7). These results suggest that oxoammonium ions may induce decomposition of thin films.
4.Please check that if the text (line 170) on page 5 is complete. “When the thin film is dried, the frequency change of the probe can be obtained fro
→Added missing text (page 5).
Reviewer 2 Report
the paper is good but some should be improved before got accepted in the journal
please the authors should rewrite the abstract to make it more attractive for the readers
please point out the novelty of their research in the introduction
figure 3 is not clear please change it
compare the results of AFM with studies from the literature
shortens the sentences in conclusion then rewrite it and be concise and precise
Author Response
Thank you for your kind suggestion of revision of our manuscript (polymers- 2056304). We have revised the manuscript according to reviewer’s comments. All revisions made are marked in red in the revised manuscript. Our responses are as follows.
1. please the authors should rewrite the abstract to make it more attractive for the readers
→Revised abstract.
2. please point out the novelty of their research in the introduction
→Added text (in red) to the introduction. We investigated whether the degradability of thin films differs depending on the functional groups of various amines and differences in the polymer structure.
3. figure 3 is not clear please change it
→modified chemical structure
4. compare the results of AFM with studies from the literature
→Added text to the manuscript(page 6). It is possible that this is similar to the adsorption of macromolecules in LbL films, rather than the adsorption of large aggregates.
5.shortens the sentences in conclusion then rewrite it and be concise and precise
→We simplified the conclusion and added the text (in red).
Round 2
Reviewer 1 Report
I carefully re-evaluated the revised manuscript and the author's response to the reviewer's comments. The authors have addressed the concerns raised by and manuscript quality has been increased. I recommend the publication of the manuscript in its current format.
Reviewer 2 Report
I accept for publication it has well improved